# PAIR: Pre-denosing Augmented Image Retrieval Model for Defending Adversarial Patches

## ABSTRACT

Deep neural networks are widely used in retrieval systems. However, they are notoriously vulnerable to attack. Among the various forms of adversarial attacks, the patch attack is one of the most threatening forms. This type of attack can introduce cognitive biases into the retrieval system by inserting deceptive patches into images. Despite the seriousness of this threat, there are still no well-established solutions in image retrieval systems. In this paper, we propose the Pre-denosing Augmented Image Retrieval (PAIR) model, a new approach designed to protect image retrieval systems against adversarial patch attacks. The core strategy of PAIR is to dynamically and randomly reconstruct entire images based on their semantic content. This purifies well-designed patch attacks while preserving the semantic integrity of the images. Furthermore, we present a novel training strategy that incorporates a semantic discriminator. This discriminator significantly improves PAIR's ability to capture real semantics and reconstruct images. Experiments show that PAIR significantly outperforms existing defense methods. It effectively reduces the success rate of two state-of-the-art patch attack methods to below 5%, achieving a 14% improvement over current leading methods. Moreover, in defending against other forms of attack, such as global perturbation attacks, PAIR also achieves competitive results. The codes are available at: https://anonymous.4open.science/r/PAIR-8FD2.

## CCS CONCEPTS

• **Security and privacy**; • **Computing methodologies → Computer vision**; • **Information systems → Information retrieval**;

## KEYWORDS

Image Retrieval, Adversarial Attack and Defense

## 1 INTRODUCTION

Multimedia retrieval has always been an important research topic due to the growth of data in cyberspace and the need for efficient data management. Deep neural networks are widely used in a variety of multimedia retrieval tasks, including content-based image retrieval [20, 21, 27] and text-based image retrieval [30, 35].

However, neural networks are vulnerable to attacks [6, 29]. This can lead to serious security issues. Patch attack [4, 19, 40] is one of the most threatening forms of adversarial attack that modifies pixels

*ACM MM, 2024, Melbourne, Australia*
© 2024 ACM.
ACM ISBN 978-1-4503-XXXX-X/18/06
https://doi.org/XXXXXXX.XXXXXXX

**Unpublished working draft. Not for distribution.**

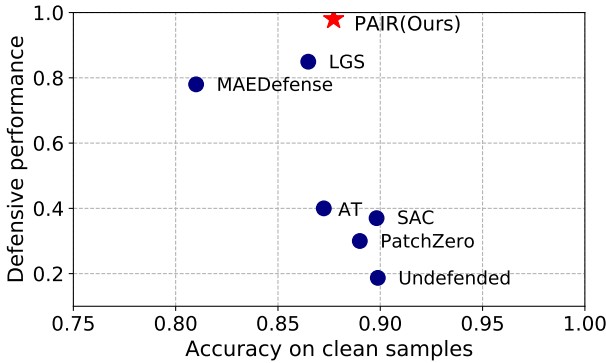

**Figure 1: Trade-off between defensive performance and retrieval accuracy on the MSCOCO. PAIR achieved the best defense and retrieval accuracy close to the original model.**

in a contiguous region of the image. Some recent studies [14, 15] have shown that the attacker can insert a patch into the image so that the users will see the search results that the attacker wants them to see. As shown in Figure 2, the attacker inputs adversarial images into a content-based image retrieval system. The search results are all irrelevant images. Such vulnerabilities could be exploited in intellectual property protection systems, allowing plagiarised content to escape detection. Moreover, the threat extends to text-based image retrieval databases, where attackers could introduce advertising images with attack patches. As a result, a user searching for a seemingly innocuous word such as "shoes" could be bombarded with these unwanted advertisements.

It is still unknown how to combat against patch attacks on content-based and text-based image retrieval systems. This problem has three main challenges: (1) Adaptation to diverse adversarial patches. Defense strategies need to be robust to the shape, form, and position of the adversarial patch. Current preprocessing defense methods [18, 37] often rely on training with specific types of adversarial images, which limits their effectiveness when encountering previously unseen patch configurations. (2) Resistance to attack without relying on localization methods. LGS [24] locates attack pixels through empirical observation, while SAC [18] and Patchzero [33] use a patch detector for localization. However, according to Chiang et al. [8] and our experiments, these localization-based defenses can be easily fooled by adaptive attacks. (3) Affordability of training and implementation. Some studies employ expensive adversarial training to defend against global perturbation attacks [38, 39]. However, in many cases, the cost of adversarial training is prohibitively expensive, particularly for models pretrained on large-scale datasets such as CLIP [26].

In this study, we propose a novel pre-denoising and purifying **defense strategy** named the Pre-denoising Augmented Image Retrieval (PAIR) model. As shown in Figure 1, PAIR achieves a superior trade-off between defense performance and retrieval ability on

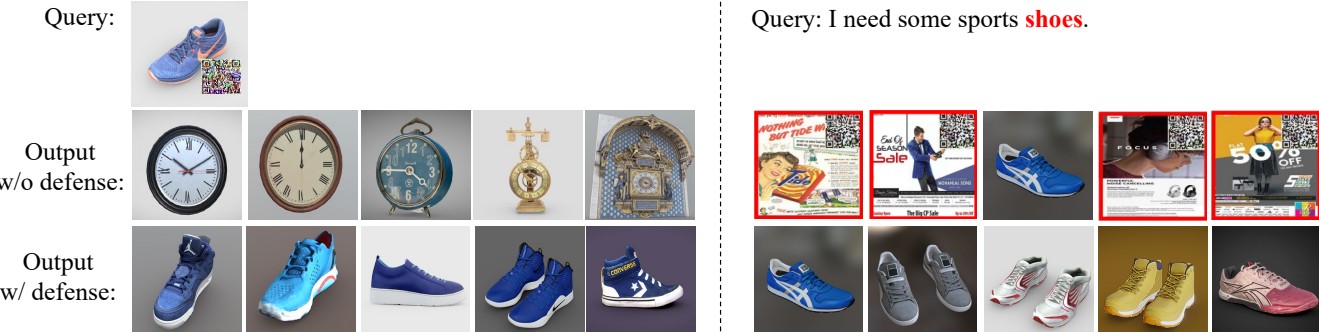

**Figure 2: Patch attack and defense scenarios. For the content-based image retrieval system, the adversarial patched image serves as the input. For the text-based image retrieval system, some advertising images with adversarial patches are added to the database. The attack patch format is set as a QR code. We develop a pre-denoising augmented strategy to protect the retrieval system from patch attacks. Above: the retrieval result without defense; Below: the retrieval result with our defense.**

clean samples. PAIR employs a unique strategy by masking random portions of image patches and reconstructing it repeatedly using a robust generative network. This approach not only guarantees that all attack patch pixels are purified, but also ensures that important semantic content is preserved. Note that as a preprocessing denoising module, PAIR is theoretically compatible with any downstream image retrieval model. Remarkably, PAIR relies solely on clean images and doesn't involve any localization techniques. This attribute makes the model relatively unaffected by variations in the shape, size, and position of attack patches. Moreover, PAIR does not require expensive adversarial training. Experiments on several datasets against patch attacks demonstrate that PAIR is highly effective in both adaptive attack and non-adaptive attack settings for content-based and text-based image retrieval.

Further, we present a new **training method** for PAIR's generative network. As a pre-denoising method, PAIR will inevitably affect the accuracy of the downstream retrieval model. This training method is designed to enhance the generative network's ability in image reconstruction, ensuring the preservation of semantic information from the original image through the implementation of a semantic discriminator. Our ablation experiments show that it can reduce the negative effects caused by PAIR. This means that the defended model has a similar performance to the original model in the case of clean samples, but with a higher defensive capability.

In summary, our main contributions are as follows:

- To the best of our knowledge, PAIR is the first defense model against patch attacks in content-based and text-based image retrieval. It is insensitive to variations in the shape, form, and position of adversarial patches.
- We design a new training method that utilizes a semantic discriminator to improve the PAIR's performance. The defense model performs similarly to the original model on clean samples and has stronger defenses.
- Experiments show that PAIR is highly effective in defending against two state-of-the-art attack methods. It reduces the success rate of attacks to less than 5%. Furthermore, PAIR also shows competitive results in defending against other forms of attack, such as global perturbation attacks.

## 2 RELATED WORK

### 2.1 Adversarial Patch Attacks

Patch attacks are one of the most threatening forms of adversarial attacks, modifying pixels in a continuous region of an image. AdvHash [15] and TTH [14] have developed patch-based attack methods in content-based and text-based image retrieval systems, respectively. AdvHash inserts an attack patch into the input image, causing the content-based image retrieval system to return irrelevant images. This attack can be used, for example, to protect intellectual property. If the plagiarised works are entered after the attack, the original works cannot be retrieved. TTH inserts adversarial advertising images into the image database of the text-based image retrieval system. After the attack, when users enter text containing specific keywords, the system displays irrelevant advertisements. Adversarial patches can be implemented in a variety of image carriers, such as logos, QR codes, text, etc., and can appear in any position in the image. They wreak havoc on content-based and text-based image retrieval systems.

### 2.2 Defenses Against Patch Attacks

**Defenses designed specifically for classification tasks.** The representative method is based on Derandomized Smoothing [7, 16, 28]. Derandomized Smoothing divides the image into several image bands. Since the number of image bands affected by the adversarial patch is limited, the final classification result can be obtained by voting according to the classification result of each image band. Besides, PatchGuard [32] designed a defense mechanism in convolutional network based on small receptive fields. However, these methods are specifically designed for image classification. It is difficult to transfer to image retrieval.

**Universal defenses against patch attacks.** Universal methods are mainly preprocessing methods, which purify and denoise the images before input to the model. Representative methods are LGS [24], PatchZero[33], SAC [18], MAEDefense [22]. LGS [24] developed a smoothing strategy based on the image gradient. It was motivated by empirical observation: the attack patch's pixel value changes dramatically. However, LGS can be easily broken

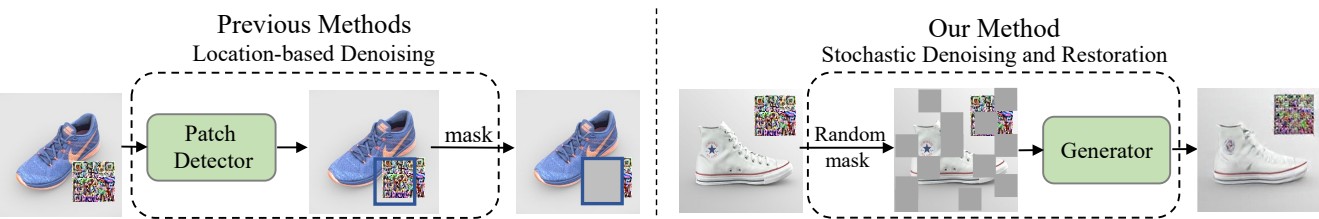

**Figure 3: Defense paradigm shift. Previous work such as SAC [18] and PatchZero [33] used a patch detector to locate and mask attack patches. This is highly dependent on the performance of the patch detector. If the patch detector is inside the attacker's target and locates it inaccurately, the localization-based defense will fail. Our defense, in contrast, randomly masks and recover the image multiple times. This mechanism guarantees that after purification, all patch pixels are reconstructed and denoised.**

by white-box attacks, as demonstrated by experiments [8], if the attacker optimizes the adversarial patch against the model that includes all pre-processing steps. SAC [18] and PatchZero [33] utilizes a patch detector to locate and mask attack patches in adversarial images. However, the clever attacker will fool the patch detector. As a result, the attack patch cannot be accurately located by the patch detector. This location-based approach is also easily broken by this adaptive attacks. MAEDefense [22] utilizes a masked auto-encoder to partially generate the image, weighted with the original image. Nevertheless, this cleaning mechanism is not thorough and not trained at the semantic level. As a result, the effectiveness of MAEDefense is limited and will have a significant negative impact on the accuracy of downstream retrieval tasks.

**Enhance robustness based on adversarial training.** Some defense methods [38, 39] attempt to conduct adversarial training (AT) on the retrieval model. This strengthens the model's resistance to adversarial samples. However, adversarial training is expensive and less practical than the preprocessing method for models that require large-scale pre-training, such as CLIP [26].

## 3 PROBLEM DESCRIPTION

### 3.1 Target Retrieval Models

• **Content-based image retrieval model.** Images are the input to content-based image retrieval systems. The retrieval system aims to find similar images in the database. We use CSQ-ResNet50 [34] as our target model in the content-based image retrieval task. The core structure of CSQ-ResNet50 is a feature extractor $H(\cdot)$. An image $x$ can be transformed into a K-bit hash code $c$ as follows:

$$c = sign(H(x)). \tag{1}$$

The Hamming distance of the images' hashcodes is used to calculate the similarity score. The similarity score is then utilized for the similarity ranking [5].

• **Text-based image retrieval model.** Texts are the input to text-based image retrieval systems. The retrieval system searches for images that match the written description. We use CLIP [26] as the model in the text-based image retrieval task. CLIP is a cross-modal model pretrained on 400 million image-text pairs. Its network structure consists of a text encoder $F_t(\cdot)$ and an image encoder $F_i(\cdot)$. Text and image representations can be obtained by feeding them into the corresponding encoder. The semantic distance $s$ between a

text $y$ and an image $x$ can be calculated as:

$$s = d(F_i(x), F_t(y)), \tag{2}$$

where $s$ is the semantic distance, which can be used to rank similarity, and its function $d(\cdot)$ selects the Euclidean distance.

### 3.2 Attack Formulation

• **Attack in content-based image retrieval systems.** AdvHash [15] is the state-of-the-art white box attack method against content-based image retrieval systems. AdvHash inserts a universal adversarial patch $\sigma$ into the input image. As a result, retrieval results may include irrelevant images of the target class. AdvHash first computes the central hash code $c_t$ of each image class. The adversarial patch is then optimized so that the hash codes of the adversarial images are close to the $c_t$.

• **Attack in text-based image retrieval systems.** TTH [14] is the state-of-the-art white box attack method for text-based image retrieval systems. TTH inserts advertising images with universal adversarial patches $\sigma$ into the image database. When a user enters some keywords, the retrieval result will include irrelevant adversarial images. TTH first computes the embedding of all sentences containing the target keyword to obtain the embedding $e_t$ associated with the target keyword. An adversarial patch is then optimized so that the embedding of the images with the attack patch is close to the $e_t$.

• **Non-adaptive attack.** In non-adaptive attacks [18], the attacker only has knowledge of the retrieval model and no knowledge of defense methods. In our experimental setup, the attacker optimizes the attack patches by gradient backpropagation based on TTH [14] and AdvHash [15] methods, targeting the original model without any defense. Then, defense methods are applied to the optimized patches. Based on the retrieval results, test their defense capabilities.

• **Adaptive attack.** Adaptive attack applies some targeted strategies [1, 2] to attack the whole model including the defense pipline. To effectively attack SAC [18] and PatchZero [33], we make fooling the patch detector one of the attacker's optimization goals. If the patch detector can't locate the patch correctly, the SAC and PatchZero defenses will fail. Since the defense pipline of LGS [24] is differentiable, attackers can directly perform gradient backpropagation to optimize the attack patch. To effectively attack PAIR, we have experimented with various adaptive attacks, such as BPDA [1] and EOT [2], as well as their combinations. Due to space limitations, we don't show the details, but we are convinced that EOT is the

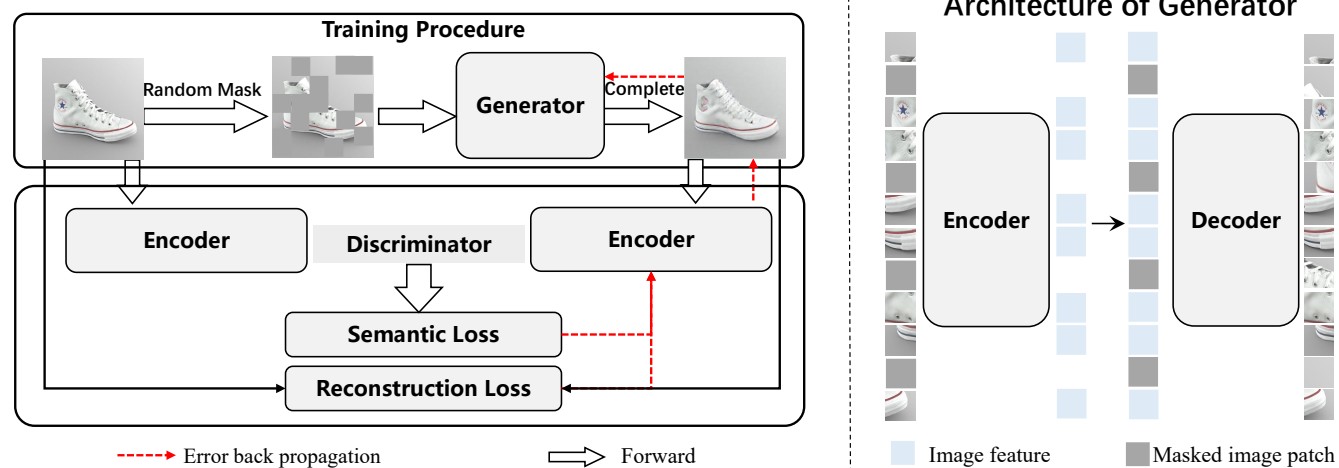

**Figure 4: Method of training the generator in PAIR. The generator consists of an encoder and a decoder. The encoder is used to extract features for each image patch. The decoder is used to infer the masked image patches. The loss function includes semantic loss at the semantic level and reconstruction loss at the pixel level.**

most effective adaptive attack strategy against PAIR. The following are the new optimized goals for TTH [14] and AdvHash [15]:

$$\arg\min_{\boldsymbol{\sigma}} \boldsymbol{d}\left(\frac{1}{h_{adv}}\sum_{h=1}^{h_{adv}} F_i(T(\boldsymbol{x}')), \boldsymbol{e_t}\right), \tag{3}$$

$$\arg\min_{\boldsymbol{\sigma}} \boldsymbol{d}\left(sign\left(\frac{1}{h_{adv}}\sum_{h=1}^{h_{adv}} H(T(\boldsymbol{x}'))\right), \boldsymbol{c_t}\right), \tag{4}$$

where $\boldsymbol{x}'$ is the attack image modified by the attacker, $T$ is the pre-processing defense method, $H$ and $F_i$ are content-based and text-based image retrieval models, $h_{adv}$ is the hyperparameter of EOT, $\boldsymbol{d}$ is the distance measure, and $\boldsymbol{\sigma}$ denotes the modification degree of the image by the attacker.

## 4 OUR MODEL PAIR

Our goal is to develop an augmented pre-denoising method that is efficient and insensitive to variations in the shape, position, and form of adversarial patches for content-based and text-based image retrieval systems.

### 4.1 Overview

In this research, we tackle the problem of defending against patch attacks. These attacks modify parts of an image to mislead retrieval models. To solve this problem, we propose PAIR (Pre-denosing Augmented Image Retrieval Model), a novel solution designed to improve the security of image retrieval systems.

The fundamental concept behind PAIR is its ability to randomly and dynamically reconstruct an entire image, following the semantic of the image. This process is achieved through a generative network in the form of a masked auto-encoder. With our designed masking and recovery mechanism, it is guaranteed that all pixels of the attack patch are purified. Such a technique ensures the preservation and capture of the semantics of the processed image, adaptable to any downstream model, while disrupting the adversarial patches.

This disruption makes the attackers' optimisation strategies less effective and harder to adapt to.

However, the traditional masked autoencoder model is insufficient to fully recover the image, which inevitably leads to a significant compromise in the accuracy of the retrieval model. To improve the recovery of PAIR, we propose a novel training methodology based on a semantic discriminator. This approach, which focuses on training the generative network from a semantic representation standpoint, markedly increases the retrieval model's accuracy. Moreover, it unexpectedly can enhance the model's ability to capture the real semantics of an image, thereby enhancing its defensive capabilities, as demonstrated in our ablation studies.

### 4.2 Pipeline of PAIR Defense

PAIR defends the retrieval model from patch attacks through dynamic and random reconstruction of the entire input image. The defence mechanism we design can be guaranteed to purify all pixels of the attack patch while preserving the semantics of the input image. The operational pipeline of PAIR unfolds through the following sequential steps:

● **Step 1:** Initially, partition an image $\boldsymbol{x} \in R^{c \times h \times w}$ into $n$ image patches $\{x_i\}_{i=1}^{n}$, each of size $s \times s$ pixels, where $x_i \in R^{c \times s \times s}$. The dimensions $c$, $h$, and $w$ represent the number of channels, height, and width of the image, respectively. Ensure that $s$ is a common divisor of both $h$ and $w$, and the total number of patches $n$ is $hw/s^2$. Subsequently, generate a binary mask sequence $M$ of length $n$:

$$M = [m_1, m_2, ..., m_n], \tag{5}$$

where $m_i$ is defined by:

$$m_i \triangleq \begin{cases} 1 & rand(0,1) > \alpha \\ 0 & \text{otherwise,} \end{cases} \tag{6}$$

where $\alpha$ denotes the mask ratio, which ranges between 0 and 1. Experimental results indicate that setting it between 0.2 and 0.8 yields satisfactory defensive effects. However, we recommend setting it

to 0.5 to achieve optimal retrieval accuracy on clean samples. Due to the space limitation, we won't show this sensitivity analysis.

• **Step 2:** Employ the mask sequence $M$ to obtain a set of masked image patches $\{p_i\}_{i=1}^n$, where $p_i$ is determined by:

$$p_i \triangleq \begin{cases} x_i & m_i = 1 \\ z & \text{otherwise,} \end{cases} \qquad (7)$$

where $z$ representing an $c \times s \times s$ image patch of zero pixel values. A generator then reconstructs the image. As depicted in Figure 4, the generator $G(\cdot)$, structured on a masked autoencoder [12], encompasses both an encoder and a decoder, each utilizing a multi-layer transformer architecture. Initiate the image restoration by mapping the image patches to embedding via a fully connected layer $\mathbf{FC}(\cdot)$, as with ViT [10]. Then, add 1-dimensional positional embedding $e_{pos}$ to the image patch embedding in raster order, following the method in [10]:

$$e_i = \mathbf{FC}(p_i) + e_{pos}. \qquad (8)$$

Here, $d$ represents the dimension of each embedding and subsequent feature. Input the embedding set $\{e_i\}_{i=1}^n$ into the encoder to infer the feature set $\{f_i\}_{i=1}^n$:

$$f_1, f_2, \ldots, f_n = \mathbf{Encoder}(e_1, e_2, \ldots, e_n). \qquad (9)$$

For masked image patches, introduce a trainable mask embedding $e_{mask}$ as a feature and combine it with 1-dimensional positional embedding. The resulting feature set $\{f_i\}_{i=1}^n$ and the mask embedding $e_{mask}$ are then input into the decoder along with 1-dimensional positional embedding $e_{pos}$:

$$q_i \triangleq \begin{cases} f_i + e_{pos} & m_i = 1 \\ e_{mask} + e_{pos} & \text{otherwise,} \end{cases} \qquad (10)$$

$$x_1', x_2', \ldots, x_n' = \mathbf{Decoder}(q_1, q_2, \ldots, q_n). \qquad (11)$$

Concatenate the restored image patches $[x_1', x_2', \ldots, x_n']$ into a new, complete image:

$$x' = \langle x_1', x_2', \ldots, x_n' \rangle, \qquad (12)$$

where $\langle \cdot \rangle$ denotes the concatenation function, sequentially merging the image patches.

• **Step 3:** Finally, perform a second restoration using the image patches reconstructed in Step 2. This ensures that all pixels in the final image are restored in alignment with the intended semantics, rather than derived from the original image. To do this, invert the mask sequence from Step 1, $[m_1, m_2, ..., m_n]$, as follows:

$$m_i' \triangleq \begin{cases} 1 & m_i = 0 \\ 0 & \text{otherwise.} \end{cases} \qquad (13)$$

Then, replicate the image restoration process from Step 2 using the new mask sequence $[m_1', m_2', ..., m_n']$ and the same generator, but this time input the image restored in Step 2.

## 4.3 Training of Generator

In the process of training a generative model, the integration of a semantic discriminator is vital. This discriminator, structured as a multi-layer transformer network, can extract semantic features from images. The generator is trained purely on clean images to guarantee that the model is not biased against attack patches. As shown in Figure 4, unlike the defensive pipeline, the training of the generator is generated only once. The motivation is that, from

our observation, iterative generation based on already recovered images will result in training instability.

To enhance the semantic consistency between the reconstructed image $x'$ and its original $x$, we have formulated a composite training loss function $L$ as follows:

$$L = L_p + \beta L_s, \qquad (14)$$

where $L_p$ denotes the pixel-level reconstruction loss and $L_s$ denotes the semantic reconstruction loss, with the parameter $\beta$ modulating their respective contributions. The pixel-level loss $L_p$, which provides hard labels for training, is formally defined as:

$$L_p = \frac{1}{n} \sum_{i=1}^n [1 - \mathbf{SSIM}(x_i, x_i')]. \qquad (15)$$

Here $\mathbf{SSIM}(\cdot)$ [31] is a metric for measuring the similarity of two images. It takes into account the brightness, contrast and structural information of the images. It yields values in the range of -1 to 1, with higher values being more similar. The following is the formula for $\mathbf{SSIM}(\cdot)$:

$$\mathbf{SSIM}(x_i, x_i') = \frac{(2\mu_{x_i}\mu_{x_i'} + c_1)(2\sigma_{x_i x_i'} + c_2)}{(\mu_{x_i}^2 + \mu_{x_i'}^2 + c_1)(\sigma_{x_i}^2 + \sigma_{x_i'}^2 + c_2)}, \qquad (16)$$

where $\mu_{x_i}$ is the average pixel value, $\sigma_{x_i}^2$ is the variance, $\sigma_{x_i x_i'}$ is the covariance of $x_i$ and $x_i'$. $c_1$ and $c_2$ are constants, following [31].

The loss $L_s$ at the semantic level provides soft labels and is formally defined as:

$$L_s = 1 - \frac{D(x) \cdot D(x')}{\| D(x) \| \| D(x') \|}. \qquad (17)$$

In this equation, the semantic discriminator $D(\cdot)$ plays a crucial role in extracting the semantic vector of the image, employing cosine similarity to maximize semantic alignment.

## 5 EXPERIMENTS

In this section, we introduce the benchmark datasets and analyze the results of defending against state-of-the-art adversarial patch attacks in the content-based image retrieval task and the text-based image retrieval task. In addition, we perform ablation studies to investigate the factors that affect defense and clean sample accuracy. Finally, we investigate the sensitivity of PAIR to the size, shape, form, and position of attack patches.

## 5.1 Experimental Setting

**Datasets.** We demonstrate the effectiveness of our approach on three popular benchmark datasets. ImageNet [9] and MSCOCO [17] are used for the content-based image retrieval task. Flickr30K [25] and MSCOCO are used for the text-based image retrieval task. For ImageNet, following [3, 15], we use a subset that has 130k images with 100 classes. We apply the COCO2014 dataset for MSCOCO. The training set contains 82,783 images and the validation set contains 40,504 images with a total of 80 categories, and each image has 5 sentences of annotation. Flickr30k contains 31,783 images, each with 5 sentences of annotation. We randomly selected 21,783 images for the training set, 5,000 images for the validation set, and 5,000 images for the test set.

**Table 1: Compared with other methods in text-based image retrieval systems. R@10 (%) under non-adaptive and adaptive attacks. The detailed attack formulation are presented in Section 3.2. "↑" means higher is better. "↓" means lower is better. The best performance of each column is in bold.**

| Dataset | Methods | Clean↑ | Adaptive Attack | | Non-adaptive Attack | |
|---|---|---|---|---|---|---|
| | | | irrelevant image↓ | Relevant image↑ | irrelevant image ↓ | Relevant image↑ |
| Flickr30K | Undefended | **91.72** | 94.01 | 71.73 | 94.01 | 71.73 |
| | PatchZero [33] | 91.70 | 79.46 | 83.82 | 27.25 | 88.47 |
| | SAC [18] | 91.72 | 89.67 | 80.39 | 25.99 | 90.38 |
| | MAEDefense [22] | 81.21 | 22.71 | 83.72 | 5.14 | 84.32 |
| | LGS [24] | 85.78 | 36.55 | 87.56 | **0.02** | 88.93 |
| | AT [23] | 88.60 | 73.99 | 86.71 | 73.99 | 86.71 |
| | Ours | 88.86 | **2.52** | **92.93** | 0.07 | **93.02** |
| COCO | Undefended | **89.88** | 81.30 | 56.42 | 81.30 | 56.42 |
| | PatchZero [33] | 89.84 | 77.91 | 61.36 | 31.34 | 74.69 |
| | SAC [18] | 89.82 | 74.42 | 62.13 | 27.75 | 75.51 |
| | MAEDefense [22] | 79.48 | 19.21 | 74.71 | 3.72 | 66.56 |
| | LGS [24] | 86.48 | 15.06 | 71.09 | 0.05 | 73.30 |
| | AT [23] | 87.24 | 71.18 | 70.30 | 71.18 | 70.30 |
| | Ours | 87.72 | **1.84** | **72.10** | **0.04** | **76.61** |

**Table 2: Compared with other methods in content-based image retrieval systems. mAP (%) under non-adaptive and adaptive attacks. The detailed attack formulation are presented in Section 3.2. "↑" means higher is better. "↓" means lower is better. The best performance of each column is in bold.**

| Dataset | Methods | Clean↑ | Adaptive Attack | | Non-adaptive Attack | |
|---|---|---|---|---|---|---|
| | | | irrelevant image↓ | Relevant image↑ | irrelevant image ↓ | Relevant image↑ |
| ImageNet | Undefended | **88.16** | 99.65 | 0.60 | 99.65 | 0.60 |
| | PatchZero [33] | 80.54 | 54.49 | 47.31 | 16.15 | 67.33 |
| | SAC [18] | 78.35 | 48.20 | 50.03 | 12.70 | 70.96 |
| | MAEDefense [22] | 65.32 | 13.47 | 76.28 | 4.85 | 78.49 |
| | LGS [24] | 74.77 | 18.41 | 79.33 | 0.46 | 85.60 |
| | AT [23] | 75.30 | 89.56 | 13.42 | 89.56 | 13.42 |
| | Ours | 76.15 | **2.05** | **84.86** | **0.21** | **87.83** |
| COCO | Undefended | **87.48** | 40.45 | 12.89 | 40.45 | 12.89 |
| | PatchZero [33] | 79.91 | 24.46 | 32.10 | 8.57 | 53.02 |
| | SAC [18] | 82.40 | 22.92 | 35.74 | 10.84 | 55.67 |
| | MAEDefense [22] | 68.24 | 7.28 | 30.85 | 3.36 | 46.92 |
| | LGS [24] | 73.42 | 2.21 | 29.96 | 0.73 | 54.34 |
| | AT [23] | 77.87 | 36.24 | 18.78 | 36.24 | 18.78 |
| | Ours | 74.72 | **1.07** | **43.42** | **0.45** | **56.28** |

**Training details.** The image patches that divide the image are $16 \times 16$ in size. The generator and discriminator are initialized with the parameters of the pretrained model MAE [12] and CLIP[26], respectively. Since CLIP is extensively pre-trained and has high generalisation capacity, the discriminator requires no further training and the generator is trained solely. A total of 104,566 images from MSCOCO and Flickr30k are used to train the generator according to the training approach. The mask radio $\alpha$ in Step 1 is set to 0.5. The $\beta$ in equation 14 is set to 1. The $c_1, c_2$ in equation 3 are set to 1e-4, and 9e-4, respectively. The batch size is set to 32, the optimizer is Adam, and the learning rate is set to 1e-5. The model is trained for 100 epochs. It takes about 20 hours to complete the training process on a Tesla V100.

**Baselines.** To compare with PAIR, we implemented PatchZero [33], SAC [18], LGS [24], MAEDefense [22], Adversarial Training (AT) [23]. In particular, we changed the uniform mask in MAEDefense's mechanism to the random mask to enhance its defense.

**Attacks details.** In the text-based image retrieval task, following [14], 23 keywords were set for target attacks. Insert 10 advertising images into the image database for each keyword. The width and height of the adversarial patch was set to 0.35 of the width and height of the original image. In the content-based image retrieval task, following [15], we tried some categories for target attacks and extracted 50 images to optimize a universal adversarial patch. Then, tested the adversarial patch on another 100 images. The length of the hashcode is set to 64. The width and height of the adversarial

**Table 3: Ablation study. R@10 (%) under different defense components in text-based image retrieval systems. "↑" means higher is better. "↓" means lower is better. The best performance of each column is in bold. The attack method is TTH [14].**

| Methods | Clean↑ | Adaptive Attack | | Non-adaptive Attack | |
|---|---|---|---|---|---|
| | | irrelevant image↓ | relevant image↑ | irrelevant image↓ | relevant image↑ |
| Undefended model | **91.72** | 94.01 | 71.73 | 94.01 | 71.73 |
| w/ reconstruction | 90.88 | 85.34 | 65.16 | 13.67 | 92.27 |
| w/ fixed mask and reconstruction | 75.06 | 85.36 | 65.06 | 0.13 | 81.04 |
| w/ random mask and reconstruction | 75.40 | 14.00 | 79.45 | 0.13 | 82.94 |
| w/ semantic discriminator (Ours) | 88.86 | **2.52** | **92.93** | **0.07** | **93.02** |

patch were set to 0.22 of the width and height of the original image. Following EBM [13], $h_{adv}$ is set to 7.

**Evaluation metrics.** We use Mean Average Precision (mAP) and Recall@K as evaluation metrics in content-based and text-based image retrieval, respectively. Similar to AdvHash [15] and TTH [14], we adopt the following three variants:

- **Clean:** test retrieval ability on clean images. To measure the negative impact of these defense methods on the model retrieval capability. The higher this metric is, the better.
- **Irrelevant image:** After the attack, irrelevant images that the attacker wants the user to see will appear in the retrieval results. For Recall@K, a successful attack is defined as an irrelevant image ranking in the top $K$. Therefore, for defensive methods, the lower this metric is, the better.
- **Relevant image:** After the attack, the ranking of really relevant images in the results will drop. This metric use ground-truth images to compute Recall@K and mAP.

## 5.2 Defense Results

We compared PAIR with several state-of-the-art defense methods from similar tasks. The detailed attack formulation is described in Section 3.2. The experimental results highlight the advantages of PAIR in the context of adaptive attacks.

**Text-based image retrieval.** As shown in Table 1, PAIR demonstrates superior defense capabilities on both datasets. In particular, against adaptive attacks, the success rate of the attacks is 14.05% lower than the second-best method. Against non-adaptive attacks, methods such as MAEDefense [22], LGS [24], and PAIR prove highly effective, reducing the attack success rate to below 1%. While SAC [18] and PatchZero [33] perform not well. If the patch detectors are not located correctly, their defence fails. PatchZero and SAC have more advantages on clean samples due to less image damage. However, PAIR also maintains a high level performance for retrieve clean samples, paralleling the original model.

**Content-based image retrieval.** The results in Table 2 highlight PAIR's strength in dealing with adaptive attacks, reducing the success rate of such attacks to below 2.05%. This level of effectiveness is also seen in the context of non-adaptive attacks. However, in terms of retrieval performance on clean samples, PAIR shows a notable decrease compared to the undefended model. On the one hand, mAP is a more comprehensive evaluation of the ranking results, reflecting the negative impact of the defense. On the other hand, this decrease is a trade-off for its much stronger defense, which results in PAIR still being the best performing method overall.

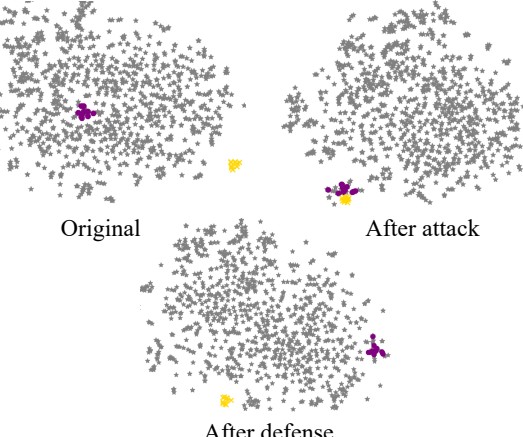

Original                    After attack

After defense

**Figure 5: High dimensional vector visualization. We extract both image and text as high-dimensional vectors and visualize them. Experimental results show that after PAIR defense, the semantic distance between the vectors is restored to the original state. Purple: texts containing the attacked keyword "policeman"; Gray: natural images in the test set; Gold: attack images inserted into the database. The goal of the attacker is to make the gold points close to the purple points.**

## 5.3 Ablation Study

In this section, we evaluate the effectiveness of the various components of PAIR. As shown in Table 3, when the image is reconstructed without masking, it has a certain defensive effect under a non-adaptive attack. In this experimental setting, the generator's encoder is employed to extract the image features. Then, the decoder is used to restore the image features to the reconstructed image. When reconstructing an image with a fixed mask, it can provide an effective defense against a non-adaptive attack. However, it provides no significant barrier against the adaptive attack.

PAIR can achieve excellent defense when reconstructing the input image using random masking under both adaptive and non-adaptive attacks. We believe that random masks will result in random damage to attack patches and that the existing adaptive attack methods are difficult to adapt. Therefore, it has an efficient defense effect under adaptive defense. The ability of the generator to preserve image semantics improves after the addition of a semantic discriminator to its training. Therefore, the retrieval accuracy on clean images is comparable to the original model's effect. The reasons for the stronger defensive effect are as follows: because the

**Table 4: Robustness Analysis. R@10 (%) under adversarial patches attack in the text-based image retrieval system. The size of the image is 224 × 224. Adaptive Attack and Non-adaptive Attack are presented in Section 3.2. "↑" means higher is better. "↓" means lower is better. Experiments show that PAIR has sufficient defense against patches of different sizes, shapes and forms.**

| patchsize | Undefended | | Our defense | | | |
| --- | --- | --- | --- | --- | --- | --- |
| | White-box Attack | | Adaptive Attack | | Non-adaptive Attack | |
| | irrelevant image↓ | relevant image↑ | irrelevant image↓ | relevant image↑ | irrelevant image↓ | relevant image↑ |
| 48×48 | 61.94 | 92.08 | 0.27 | 91.72 | 0.04 | 93.25 |
| 80×80 | 97.84 | 62.07 | 2.85 | 93.91 | 0.03 | 92.88 |
| 112×112 | 99.54 | 37.35 | 4.57 | 93.55 | 0.02 | 92.52 |
| rectangle (40×160) | 90.72 | 77.64 | 3.92 | 93.03 | 0.03 | 93.46 |
| circle (radius=45) | 97.29 | 34.21 | 3.53 | 93.24 | 0.11 | 92.38 |
| Other forms and positions (80×80) | 92.59 | 72.10 | 4.87 | 91.43 | 0.07 | 94.40 |

ground-truth image is better reconstructed, its retrieval ranking will be higher. As a result, the metric of the relevant image improved. With the enhanced ability of the model to capture real semantics, the ability to purify against adversarial perturbations is increased. Therefore, it benefits the metric of the irrelevant images.

## 5.4 Robustness Analysis

We evaluated the robustness of PAIR against adversarial attacks of different patch sizes, shapes, and forms. The results in Table 4 demonstrate that our proposed PAIR approach maintains high effectiveness against attacks of varying sizes. As the size of the adversarial patches increases, the attack success rate generally increases when no defense is applied. However, PAIR consistently demonstrates high efficiency in defense, with attack success rates of less than 5%. Then, we conducted experiments with different patch shapes, including circles and rectangles. Furthermore, we randomly altered the form and position of the attack patches. As shown in Table 4, we reported the average experimental data. The experimental results show that PAIR remains effective. This is attributed to PAIR's reliance solely on clean samples for training, making it insensitive to diverse attack patches.

**Visualization.** The visualization is designed to confirm that our defense lowered the aggression of the attack patch. The attacks on PAIR are set to be adaptive attacks to achieve better attack effects. As shown in Figure 5, we visualized the semantic representation of the original images and texts. After the PAIR defense, the semantic distance between the attacker's images and the texts containing the attacked keyword is restored to its original state.

## 5.5 Defense against Global Perturbation Attacks

To illustrate the generality of PAIR, we experiment against global perturbation attacks. The global perturbation attack allows the attacker to modify all the pixels in the image and set the perturbation individually for each adversarial image. The attack is set up to insert an adversarial advertising image into the database. Modify the TTH [14] to the global perturbation attack on the MSCOCO [17] dataset. The size of perturbation budget is 4/255 in $l_\infty$. Partial defense methods can't defend against global perturbation attacks, such as PatchZero [33] and SAC [18]. Therefore, we introduce other advanced baselines for comparison. Among them, JPEG [11], ARN [36], and CAFD [37] are based on pre-processing denoising and HM [39] is based on adversarial training. As shown in Table 5, compared

**Table 5: Comparison with other aviliable methods in defense against global perturbation attacks. R@10 (%) under adaptive attacks in text-based image retrieval systems.**

| Method | irrelevant image ↓ | relevant image ↑ |
| --- | --- | --- |
| Undefended | 99.28 | **79.56** |
| ARN [36] | 96.47 | 76.57 |
| CAFD [37] | 96.24 | 75.63 |
| MAEDefense [22] | 59.42 | 64.86 |
| JPEG [11] | 47.26 | 55.65 |
| HM [39] | 48.82 | 68.65 |
| Ours | **42.29** | 75.42 |

to other preprocessing defense methods, PAIR can achieve better defense. Due to less damage to the image, methods such as CAFD and ARN perform better on the metrics of a relevant image, but offer little defense against adaptive attacks.

## 6 CONCLUSIONS

In this paper, we have developed PAIR, a new method specifically designed to protect against patch attacks in both content-based and text-based image retrieval systems. This method processes images in a way that reduces the impact of attacks while preserving the semantic information for the downstream model. Extensive experimental evaluations on a variety of benchmark datasets has demonstrated PAIR's exceptional ability to defend against adversarial patch attacks. Impressively, PAIR is insensitive to the shape, form, and position of the attack patches. Furthermore, in terms of retrieval accuracy on clean samples, PAIR's performance is comparable to that of the original, undefended model. Ablation studies further reveal that the PAIR module introduces unpredictable disruptions to attack patches, making existing adaptive attack methods less effective and difficult to adapt to.

**Limitations and broader impacts.** Although our PAIR defense performs well in semantic-based retrieval models, it is not suitable for fine-grained retrieval tasks, such as face retrieval. In terms of compatibility, as a preprocessing defense, our method destroys the attack patches while preserving the image's semantic information, making it compatible with the most of downstream retrieval models. Moreover, our defense strategy may be extended to other domains, such as defense in graphs and texts.

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
