# OpenReview forum: "PAIR: Pre-denosing Augmented Image Retrieval Model for Defending Adversarial Patches"
_acmmm.org/ACMMM/2024/Conference — MM2024 Poster_

### Official Review · Reviewer_ZnJG · 2024-05-22

**Rating:** 3
**Confidence:** 3

**Summary:**

This paper proposes the Pre-denosing Augmented Image Retrieval (PAIR) model, a new approach designed to protect image retrieval systems against adversarial patch attacks. Experiments show that PAIR significantly outperforms existing defense methods. It effectively reduces the success rate of two state-of-the-art patch attack methods to below 5%, achieving a 14% improvement over current leading methods

**Strengths:**

(1) PAIR is the first defense model against patch attacks in content-based and text-based image retrieval. It is insensitive to variations in the shape, form, and position of adversarial patches
(2) PAIR is highly effective in defending against two state-of-the-art attack methods. It reduces the success rate of attacks to less than 5%. Furthermore, PAIR also shows competitive results in defending against other forms of attack, such as global perturbation attacks.
(3) The writing of this paper is well and can be easy to follow.

**Limitations:**

(1) The novelty of this paper is limited. There are too many studies using the generative model to defend the adversarial attacks and backdoor attacks of neural networks. The proposed method is straightforward.
(2) Why only consider the adversarial patch? What is the performance of the proposed method against original adversarial attacks of the Image Retrieval Model?
(3) The proposed method needs to train a generative model. Whether using pre-trained generative models such as stable diffusion can also effectively against adversarial attacks

**Suitability:**

3

---

### Official Review · Reviewer_wyiF · 2024-05-22

**Rating:** 4
**Confidence:** 2

**Summary:**

The paper introduces the PAIR model to defend image retrieval systems against patch attacks. PAIR dynamically reconstructs images based on their content, thwarting patch attacks while maintaining semantic integrity. It incorporates a semantic discriminator in training to enhance semantic understanding and reconstruction accuracy. Experimental results demonstrate PAIR's effectiveness.

**Strengths:**

- The Mask-reconstruct approach is a certain level of innovation.
- The method is straightforward and easy to implement, with thorough experimentation.
- The code has been released.

**Limitations:**

-	In line 300, it is mentioned that "Previous work such as SAC and PatchZero used a patch detector to locate and mask attack patches. This is highly dependent on the performance of the patch detector." This statement similarly applies to this work, as the PAIR method is also highly dependent on the performance of the generator. If the generator is not sufficiently good, the reconstructed image quality will suffer, potentially resulting in significant structural differences from the original image.

-	There is a substantial risk associated with masking the original image and then reconstructing it. If the image contains text or human faces, for example, it would be challenging to recover details after masking, which could significantly impact the image database negatively.

-	Experiments lacking efficiency comparison. If the reconstruction process consumes a considerable amount of resources, it may not be an ideal defense method.

Please update the TTH citation to reference [1].

[1] Hu, Fan, Aozhu Chen, and Xirong Li. "Towards Making a Trojan-Horse Attack on Text-to-Image Retrieval." ICASSP 2023-2023 IEEE International Conference on Acoustics, Speech and Signal Processing (ICASSP), 2023.

**Suitability:**

3

---

### Official Review · Reviewer_g5Wu · 2024-06-02

**Rating:** 4
**Confidence:** 3

**Summary:**

The paper proposed the Pre-denosing Augmented Image Retrieval (PAIR) model for defending image retrieval systems against adversarial patch attacks. The method is based on the idea of masked autoencoders that randomly masks out and reconstructs the entire images in order to remove the adversarial patches. To improve the reconstruction performance, the paper proposed to include the SSIM loss as an additional regularization term for training the generator. Experiment results show that the proposed method is highly effective in defending two state-of-the-art attack methods.

**Strengths:**

1. The paper is in general well written and easy to follow.
2. The proposed method is general and can be applied to any image retrieval system.
3. The paper conducted extensive evaluations on both content-based and text-based image retrieval systems with both adaptive and nonadaptive attacks. The proposed method demonstrates superior performance compared to the baselines.

**Limitations:**

1. The novelty of the proposed method seems incremental compared to MAEDefense [22]. It seems the main difference is the added SSIM regularization term for training the generator.
2. The random masking scheme might cause information loss that cannot be recovered by the generator.
3. The performance of PatchZero and SAC seems to be too low. Since the attack patch format is set as a QR code, it should be very easy to be detected by the patch detectors. Have the authors retrained the patch detectors on the dataset?
4. For the text-based image retrieval scenarios, it seems the defense needs to reconstruct every image in the database, which can be computationally expensive.
5. How is the reconstruction quality? Can the authors provide some visualization of before and after defense images?

**Suitability:**

3

---

### Official Review · Reviewer_W18u · 2024-06-11

**Rating:** 4
**Confidence:** 3

**Summary:**

The paper introduces the Pre-denoising Augmented Image Retrieval (PAIR) model, aimed at defending against adversarial patch attacks in image retrieval systems. The approach involves dynamically reconstructing images based on their semantic content to purify adversarial patches while maintaining the semantic integrity of the images. The authors also propose a novel training strategy incorporating a semantic discriminator to enhance the model's performance. Experimental results demonstrate that PAIR significantly reduces the success rate of state-of-the-art patch attack methods and shows competitive results against other forms of attacks.

**Strengths:**

(1)	The PAIR model introduces a novel method for defending against patch attacks by reconstructing images based on their semantic content.
(2)	The experimental results are robust, showing that PAIR reduces the success rate of patch attacks to below 5%, which is a notable improvement over existing methods.
(3)	PAIR's preprocessing denoising module is theoretically compatible with any downstream image retrieval model, making it a versatile solution that can be applied broadly across different systems.
(4)	The authors provide source code to prompt reproducibility.

**Limitations:**

(1)	The implementation of the PAIR model involves a sophisticated generative network and a semantic discriminator. This complexity might pose challenges for practical deployment in real-world systems.
(2)	While the paper presents strong results against specific patch attack methods, it would be beneficial to see a broader evaluation against a wider range of adversarial patch attack methods to fully establish the model's robustness.
(3)	As shown in Table 1, the proposed method performs slightly worse than LGS under non-adaptive attack.
(4)	The performance of the proposed method on clean images is comparatively lower than the baselines and the undefended model.

**Suitability:**

2

---

### Meta-Review · Area_Chair_Hz9u · 2024-07-03

**Recommendation:** Accept (Poster)
**Confidence:** 5

**Metareview:**

This paper introduces the PAIR model, which defends image retrieval systems against patch attacks by dynamically reconstructing images based on their content. This approach thwarts patch attacks while maintaining semantic integrity. To enhance semantic understanding and reconstruction accuracy, PAIR incorporates a semantic discriminator in training. Experimental results demonstrate the effectiveness of PAIR.

During the review process, this paper received split final reviews (2 borderline reject and 2 borderline accept). After carefully reviewing the paper and considering the reviewers' comments, as well as discussing it with the SAC, I recommend accepting this paper. The reason for my recommendation is that it presents a simple yet effective method that provides valuable insights into defending against adversarial patches.

One main concern raised by reviewers who gave negative scores is the high computing cost associated with PAIR. This concern is valid, but I believe that future academic and engineering development can address these issues of computing cost and efficiency effectively. I encourage the author to incorporate the discussions and comments from the reviewers into the paper. Additionally, it would be beneficial to improve the proposed method to reduce its computing cost in further study.